# Huperzine—A Improved Animal Behavior in Cuprizone-Induced Mouse Model by Alleviating Demyelination and Neuroinflammation

**DOI:** 10.3390/ijms232416182

**Published:** 2022-12-19

**Authors:** Hongyu Zhang, Danjie Wang, Jingxian Sun, Yumeng Wang, Shuai Wu, Jun Wang

**Affiliations:** 1Department of Integrative Medicine and Neurobiology, School of Basic Medical Science, Institutes of Integrative Medicine, Shanghai Medical College, Fudan University, Shanghai 200032, China; 2Department of Integrative Medicine and Neurobiology, State Key Laboratory of Medical Neurobiology and MOE Frontiers Center for Brain Science, Institutes of Brain Science, Shanghai Medical College, Fudan University, Shanghai 200032, China; 3Department of Neurology, Zhongshan Hospital, Shanghai Medical College, Fudan University, Shanghai 200032, China

**Keywords:** multiple sclerosis, cuprizone model, huperzine A, microglia, oligodendrocyte precursor cell

## Abstract

Huperzine A (HupA) is a natural acetylcholinesterase inhibitor (AChEI) with the advantages of high efficiency, selectivity as well as reversibility and can exhibit significant therapeutic effects against certain neurodegenerative diseases. It is also beneficial in reducing the neurological impairment and neuroinflammation of experimental autoimmune encephalomyelitis (EAE), a classic model for multiple sclerosis (MS). However, whether HupA can directly regulate oligodendrocyte differentiation and maturation and promote remyelination has not been investigated previously. In this study, we have analyzed the potential protective effects of HupA on the demylination model of MS induced by cuprizone (CPZ). It was found that HupA significantly attenuated anxiety-like behavior, as well as augmented motor and cognitive functions in CPZ mice. It also decreased demyelination and axonal injury in CPZ mice. Moreover, in CPZ mice, HupA increased mRNA levels of the various anti-inflammatory cytokines (*Arg1*, *CD206*) while reducing the levels of different pro-inflammatory cytokines (*iNOS*, *IL-1β*, *IL-18*, *CD16,* and *TNF-α*). Mecamylamine, a nicotinic acetylcholinergic receptor antagonist, could effectively reverse the effects of HupA. Therefore, we concluded that HupA primarily exerts its therapeutic effects on multiple sclerosis through alleviating demyelination and neuroinflammation.

## 1. Introduction

Multiple sclerosis (MS) is a kind of autoimmune neurological disease caused by an autoimmune attack on central nervous system (CNS) white matter [1]. It’s accompanied by immune cell infiltration, demyelination, and axonal degeneration [2], which usually can lead to the gradual decline of motor function or even disability. MS lesions are pathologically classified into four distinct patterns (I-IV) such as complement activation, IgG deposition, and myelin-associated glycoprotein loss [3,4]. Spontaneous remyelination occurs after demyelination but is often not completed [5,6]. It has been reported that more than 2 million people suffer from MS in the world, including 400,000 in the USA alone [7]. Therefore, the analysis of the causes of myelin reconstruction failure and the study of myelin repair methods are the challenges of MS research. The study included the use of animal models to help understand the mechanisms of demyelination and remyelination. Although several new disease-modifying drugs have been found to be beneficial in the treatment of MS, more novel treatment strategies that can target both neuroprotection and remyelination are required in the future [8].

Acetylcholine hydrolysis (ACh) can be regulated by AChEI, which determines the cholinergic tone and cholinergic blockade in the inflammatory process [9]. HupA, a kind of alkaloid obtained from Huperzia serrata [10], can act as a mixed-competitive, invertible, and selectable AChEI [11,12,13]. In traditional Chinese medical science, it was extensively applied to treat schizophrenia, inflammation, pain, and memory loss [14,15,16]. Studies have also indicated that HupA treatment may be more beneficial to dementia [11] and epilepsy [17] than psychotherapy and conventional drugs. Meanwhile, HupA can significantly improve the cognitive ability of Alzheimer’s disease (AD) patients, and its effect is dose -and time-dependent [18]. Low doses of HupA have been shown to effectively improve cognitive dysfunction, reduce neuroinflammation, enhance neuroprotection by increasing cortical inhibition, and play a role in various animal models of neurological diseases. For instance, HupA has antiepileptic potential to alleviate seizures in the 6-Hz model and pentylenetetrazole (PTZ) rat epilepsy model [19,20]. Importantly, in D-gal-induced hepatic injury mice, HupA exhibited protective effects against hepatotoxicity and inflame-aging by inhibiting acetylcholinesterase (AChE) activity and activating cholinergic anti-inflammatory pathways [21]. Clinical studies have also shown that patients treated with HupA for a long time can exhibit substantial improvement in anxiety, depression, and other emotional problems [22]. Coincidentally, our previous findings also demonstrated that HupA could significantly alleviate the nerve function damage of EAE mice and reduce neuroinflammation [23]. However, we are still unknown whether HupA could decrease CPZ-induced demyelination and axonal injury. Moreover, in comparison to other cholinesterase inhibitors, such as donepezil, rivastigmine, and galantamine, HupA has been found to possess stronger penetrating power, higher bioavailability and can inhibit AChE for a longer time [24,25]. Taken together, HupA can be used as novel drug for MS treatment via reducing demyelination and axonal injury in demyelinating mice.

CPZ is a copper-chelating reagent that can be mixed with rodents’ normal diet. Continuous CPZ feeding leads to a pathologic pattern similar to that of MS III lesions in white matter. However, the pathological response after CPZ intoxication in C57BL/6 mice is highly reproducible and well characterized. In this model, demyelination is evident in several structures, including the hippocampus [26], cerebellum [27], striatum [28], cerebral cortex [29], and most notably, corpus callosum [30]. The CPZ model is a mature research paradigm to explore the demyelination and remyelination processes [31,32]. CPZ is unable to damage the new oligodendrocytes. Hence spontaneous remyelination occurs during and following demyelination [33]. Anyway, removing CPZ from the diet can also promote myelin regeneration [34]. In this paper, we employed the CPZ model to assess the potential effects of HupA on behavior disorder, myelin sheath, mature oligodendrocytes defects, and neuroinflammation.

## 2. Results

### 2.1. HupA Enhanced the Motor Coordination Function of Demyelinated Mice

HupA treatment began at week 4 and lasted for 2 weeks until mice were sacrificed at week 6 (Figure 1A). The weight of CPZ mice was found to be distinctly lower than the control mice after one week. However, there was no difference observed between HupA treatment and CPZ mice (Figure 1B).

CPZ has been reported to affect the motor coordination and balance of rodents [35,36]. Therefore, we used the beam walking test (BWT) and rotarod test to detect motor coordination and balance. After 2 weeks of treatment with HupA, it was noted that the mice spent less time than CPZ mice in BWT (Figure 1C). In the rotarod test, HupA treatment mice spent more dwell time on the rotarod than CPZ mice (Figure 1D). Mecamylamine, an antagonist of the nAChR, significantly abolished the therapeutic effects of HupA on CPZ mice in these two tests. Overall, these findings suggested that HupA treatment significantly improved motor coordination after 2 weeks of treatment.

### 2.2. HupA Diminished Anxiety and Ameliorates Spatial Memory Deficit in Demyelinated Mice

Anxiety-like behavior has been observed in CPZ mice [37,38]. Open field test (OFT) was assessed to appraise their motor activity and anxiety-like behavior in a novel environment. The total distance reflects the movement activity of the mice, and the center distance, duration of the movement in the center area, and entries of the center area reflect the anxiety of mice. The habituation pattern was measured for 5 min in the square (Figure 2A). Compared to CPZ mice, HupA-treated mice showed significant effects on the total distance (Figure 2B). Moreover, CPZ mice spent less time on the centre area, such as centre distance, duration of the movement in the centre area, and entries of the centre area (Figure 2C–E), thereby suggesting that CPZ mice displayed anxiety-like behaviors. However, HupA treatment effectively relieved these symptoms after 1 week of treatment (Figure 2C–E). 

CPZ could also lead to cognitive dysfunction in demyelinating lesions [39,40]. Y maze spontaneous alternation test, a remarkable model-therapy interaction, can reflect the reference and working memory of mice through arm entries and spontaneous alternation, respectively. Frequencies of arm entries were not affected by HupA (Figure 2F), indicating CPZ did not affect the learning and exploration ability of mice. However, it was found that compared with the control mice, CPZ mice showed an obvious spatial memory deficit. HupA ameliorated the spatial memory of demyelinated mice (Figure 2G). In summary, the above findings demonstrated that HupA treatment significantly improved the behavioral abnormalities in the mice. However, mecamylamine antagonized the therapeutic effects of HupA.

### 2.3. HupA Promoted the Expression of Myelin Basic Protein (MBP) in the Corpus Callosum and Increased the Quantity of the Myelinated Axons

MBP is an abundant protein in CNS myelin. MBP has long been recognized as a factor in the pathogenesis of the autoimmune neurodegenerative disease MS and is closely related to the expression of myelin [41]. Luxol fast blue (LFB) is a copper phthalide dyestuff, which has the characteristic of binding to myelin in an alcohol solution. The myelin structure of nerve tissue can be well visualized by LFB staining [42]. So, they are chosen as appropriate markers for myelinization. In order to confirm whether the improvement in behavior abnormalities is related to remyelination, western blot, MBP, and LFB staining were simultaneously performed to analyze the effect of HupA on myelin sheath during demyelination. The results suggested that the myelin sheath in the corpus callosum was evidently lost in the demyelinated mice. However, HupA could effectively increase the expression of MBP in these two assays (Figure 3A–D); LFB staining was completely consistent with the above results (Figure 3E,F), thereby indicating that HupA can exhibit protective or regenerative effects in demyelination. However, mecamylamine significantly suppressed the therapeutic effects of HupA in these three assays.

Transmission electron microscopy (TEM) imaging can be used to observe the ultrastructure of the myelin sheath and the number of myelinated axons, which is an objective standard to evaluate myelin regeneration. We used TEM to verify the effect of HupA on remyelination. It could be seen that there were many necrotic and vacuolated axon structures in the CPZ mice (Figure 3G). After 2 weeks of HupA treatment, the number of myelinated axons was found to be significantly increased (Figure 3G and Appendix A). G-ratio serves as an important standard to measure the thickness of myelin sheath. It has been negatively correlated with myelin thickness. As shown in the picture, the average myelin sheath thickness was increased markedly. The g-ratio of HupA-treated mice was significantly reduced, thus suggesting that HupA might also promote the formation of a new myelin sheath (Figure 3H). Through observation, we preliminarily concluded that HupA could effectively promote the improvement of animal behavior by alleviating demyelination.

### 2.4. HupA Did Not have an Impact on the Quantity of Oligodendrocyte Precursor Cells (OPCs) but Increased the Quantity of the Mature Oligodendrocytes In Vivo Settings

So how does HupA promote remyelination? To further figure out whether HupA regulates the remyelination via promoting the amount of OPC or OPC differentiation, we performed immunohistofluorescence (IHC)to verify. The results demonstrated that HupA did not exhibit a substantial impact on the quantity of PDGFRα^+^ Olig2^+^ positive OPCs in the corpus callosum (Figure 4A–C), thus indicating that HupA could not significantly influence the proliferation of OPCs. Next, we observed the potential effect of HupA on the mature oligodendrocytes and found that HupA can effectively increase the quantity of CC1^+^ Olig2^+^ mature oligodendrocytes in the corpus callosum (Figure 4D–F). Therefore, we initially inferred that HupA could regulate the process of remyelination by promoting OPC differentiation.

### 2.5. HupA Did Not Affect the Proliferation of OPCs but Promoted the Differentiation of OPCs In Vitro Settings

We also determined the effect of HupA by culturing OPCs in vitro. Three different concentrations (0.1/1/10 μM) of HupA did not markedly affect the quantity of PDGFRα+ OPCs in the corpus callosum by Cell Titer-Glo (CTG) (Figure 5C), which was consistent with Figure 4A–C. Due to the high sensitivity and long signal duration, CTG has become the mainstream cell viability detection method, which can quickly and sensitively detect the number of viable cells [43]. However, immunocytofluorescence (ICC) assays also suggested that HupA (0.1/1/10 μM) promoted the differentiation of OPCs in vitro settings (Figure 5A,B). Western blot results displayed that HupA induced OPC to differentiate into mature oligodendrocytes, especially when the concentration was 1 μM, which was the same as the results in 5A, B (Figure 5D,E).

### 2.6. HupA Can Regulate the Activation of Microglia in the Corpus Callosum of Demyelinated Mice

Except for OPC, the microglia were reported to be irreplaceable in the disease process of MS. A number of studies have shown that HupA may directly act on microglia so as to reduce inflammation in CNS [44]. Through IHC, we found that after HupA treatment, the number of microglia in the corpus callosum decreased significantly (Figure 6A,B), and so is in the hippocampus (Appendix A). In order to more visually illustrate the changes in microglia number, the structure and morphology of the microglia [45] were further analyzed (Figure 6C). The results exhibited that CPZ mice had the highest proportion of microglia in the corpus callosum and showed more branch ends and total branch length (Figure 6D–F). Both the number of microglia branch ends and the total branch length in the HupA-treated mice were significantly reduced, thus suggesting that HupA could effectively regulate the activation of the microglia. However, mecamylamine was observed to antagonize the therapeutic effect of HupA. This finding also revealed that HupA might act on the microglia through the classic N-type choline receptor pathway in order to improve the inflammatory microenvironment and promote the regeneration of the myelin.

### 2.7. HupA Down-Regulated and Up-Regulated the mRNA Expression of Pro-Inflammatory and Anti-Inflammatory Genes in the Microglia Respectively

It has been established that the microglia-associated mRNA of pro-inflammatory genes and anti-inflammatory genes is related to the pathogenesis of MS. To explore the potential influence of HupA on CPZ-induced inflammation, the mRNA expression of the various pro-inflammatory and anti-inflammatory genes in the microglia was analyzed by quantitative RT-PCR (qRT-PCR). The results exhibited that HupA significantly reduced the mRNA expression of pro-inflammatory microglia-related genes (*iNOS* and *CD16*) and up-regulated the mRNA expression of anti-inflammatory microglia-related genes (*Arg1* and *CD206*) (Figure 7A–D). Besides, we also observed the effect of HupA on the expression of the different inflammatory factors. The results also indicated that HupA inhibited the expression of inflammatory factors (*IL-1β*, *IL-18*, and *TNF-α*) (Figure 7E–G). Taken together, we draw the conclusions that HupA inhibited and promoted the expression of pro-inflammatory and anti-inflammatory microglia, respectively.

## 3. Discussion

The myelin sheath is primarily responsible for maintaining the morphology and survival of axons for ensuring that the nerve signals are transmitted quickly and accurately [46,47,48]. Pathological loss and the genetic defects of myelin sheath can contribute to the occurrence of various neurological diseases. Therefore, demyelination could be observed in congenital and acquired myelin diseases, such as MS, which involves the degeneration of axons and neurons [49,50,51], or even permanent neurological disorders, such as dementia [52] and schizophrenia [53]. In addition, demyelination causes chronic inflammation, and patients will experience clinical symptoms such as hypoesthesia, anxiety, or motor functions decline. Moreover, during the early stage of the disease, the brain transmits signals from the damaged area to the undamaged area by promoting the activation, proliferation as well as migration of OPCs to demyelinating lesion areas, and the function of the damaged nerve cells can be partially compensated to maintain the optimal nerve function. Although it can be repaired by inducing the remyelination in the early stage, as the damaged area increases, self-repair can no longer compensate for the damage to axons. This might lead to the partial loss of function of the central nervous system or even cause brain atrophy. A number of different immunomodulatory drugs have been used to reduce the recurrence of the disease, but this therapy has significant side effects and can display only limited repair ability towards the demyelinated part. Several studies are being conducted to identify novel strategies for promoting the regeneration of myelin.

It has been reported that 70% of MS patients display cognitive impairment in all stages and subtypes of the disease [54,55,56,57]. The specific cognitive domains affected may vary from person to person, and patients usually show deficits in both learning and working memory. AChEI can reduce the decomposition rate of ACh to increase the expression of ACh. ACh can effectively participate in the regulation of central and peripheral inflammation to improve the microenvironment in vivo. Immune cells, including astrocytes and microglia, can respond to the increase and stimulation of ACh by activating the cholinergic receptors, thereby effectively enhancing the cholinergic neurotransmission function in the patient. Among these receptors, studies have shown that the nicotinic acetylcholine receptor (nAChR) played an irreplaceable role in Alzheimer’s and Parkinson’s disease [58,59]. It is a ligand-gated ion channel expressed in key brain regions responsible for cognitive functions, such as the cerebral cortex and hippocampus. Besides, nAChR was involved in the pathogenesis of myasthenia gravis and inflammation in the EAE model [60,61,62]. HupA, a kind of AChEI, has been used to treat various symptoms such as bruises, swelling, strain, and muscle cramps. Therefore, several clinical studies have primarily focused on analyzing the effects of HupA effects on cognition and memory [8,18,63]. In addition, precious reports on the structure and pharmacological properties of HupA found that HupA could participate in anti-inflammation and neuroprotection. Then, we selected HupA as the pivotal drug in this study. The previous findings have shown that HupA significantly improved the neurological score of EAE mice and reduced the neuroinflammatory response in the spinal cord tissues [23]. Moreover, another study found that HupA attenuated the disease process by inhibiting the inflammation of the spinal cord and the proliferation of the cerebrospinal T cells [64]. These findings displayed that HupA may be beneficial against MS. However, the study did not observe the expression of pro-inflammatory or anti-inflammatory genes in the microglia, and the relationship between the effect of HupA on oligodendrocytes and myelination in the CPZ model was not analyzed.

The CPZ model is associated with low experiment cost, high success rate, and reproducibility, which can reflect the several important characteristics of MS, and is suitable for research on the demyelination and identification of the drugs for the promotion of myelination [65,66,67]. CPZ causes extensive demyelination of the corpus callosum, resulting in motor coordination disorders [35,36]. BWT and rotarod tests are the critical basis for judging the successful establishment of the CPZ model. We found that HupA treatment didn’t affect the weight of mice, but it could improve the impaired motor coordination ability in the CPZ mice. As MS patients are often depressed, anxious and forgetful, so we included two behavioral experiments, Y maze, and OFT. Y maze test showed that their spatial memory was significantly impaired [40]. HupA treatment could significantly improve the spatial working memory of mice. OFT results reflected that the mice had anxiety symptoms [68,69,70]. HupA treatment could markedly reduce anxiety in mice, which was similar to the other literature reports [25,71]. Interestingly, both experiments showed that the control group and HupA group have undifferent phenotypes statistically, and, except for OFT, the phenotypes were maintained from week 5 to week 6. It is possible to account for the phenomenon that CPZ causes mild anxiety but not the main symptom [72], and repeated open field tests will interfere with the evaluation of anxiety. This is the reason why CPZ mice improved outcomes at week 6 in OFT.

OPC is very important for the repair and regeneration of myelin. In disease states, inhibitory signals can hinder the effective differentiation of OPC and remyelination. Thus, promoting OPC differentiation and remyelination might have important scientific significance for the treatment of CNS demyelinating diseases. Among them, the development of novel drugs that can effectively promote OPC differentiation, myelin repair, and regeneration is also an important direction for neuroscience basic and clinical research. In this project, we found that HupA did not exhibit any effect on the proliferation of OPCs, but it could promote their differentiation through IHC in vivo and culture of OPCs in vitro. However, Xie’s research has previously shown that donepezil could promote remyelination, while other AChEI did not possess such effects [73]. Western blot assay suggested that 1 μM HupA had the best effect in promoting the OPCs differentiation, while the concentration used by Xie’s group is 5–40 μM, and low-dose HupA could possibly exert a powerful inhibitory effect. We speculated that the high concentration of HupA might be toxic to the cells, which might be the main reason for the inconsistent results. In addition, whether the high concentration of HupA can be toxic still needs to be verified by further research under in vitro and in vivo settings. According to the reports, mature oligodendrocytes mainly express muscarinic acetylcholine receptors (mAChR), which are primarily M3 and coupled to a variety of signal transduction pathways [74]. No immunoreactivity to the nAChR subunit was detected in cells induced to differentiate into oligodendrocytes [75]. Mecamylamine, a nicotinic acetylcholinergic receptor, was used to elucidate the mechanism of HupA on remyelination, but it did not seem to apply to OPC. Therefore, the mecamylamine condition was not present in the results, such as in Figure 4. As for the TEM analysis, the mecamylamine group was absent because results could be proved by both MBP and LFB staining.

Microglia, as mononuclear phagocytes [76], is the resident macrophages present in the brain and are located in the parenchyma of the CNS [77,78]. They contribute to maintaining tissue homeostasis and are beneficial to the pathological changes of the CNS. They can also respond to changes in the microenvironment and produce diverse reactive phenotypes with time. For instance, they can promote the phagocytosis of myelin fragments and the secretion of different growth factors and neurotrophic factors. Thereafter, the secretion of inflammatory factors, nitric oxide, and glutamate can induce the occurrence of central nervous system inflammation, thereby causing secondary damage to axons [49]. Microglia, acting as a double-edged sword for MS, have the properties of aggravating both neuroinflammation and neuroprotection. Thus, understanding the dual role of microglia is useful for understanding the progression of diseases and finding suitable therapeutic targets. Accumulated evidence shows that nAChR, which especially contains the α7 subunit, contributes to the regulation of microglial activity through the inhibition of the synthesis of proinflammatory molecules. It is a reason to believe that HupA might act on the microglia through the classic N-type choline receptor pathway in order to work. Therefore, we observed the regulatory effect of HupA on microglia in CPZ mice. Results also indicated that HupA significantly inhibited the activation of pro-inflammatory genes and the release of inflammatory factors. Mecamylamine could reverse the efficacy of HupA on microglia, which confirmed what we thought. 

Based on the principles of drug recycling and disease connectivity, HupA, a commonly used drug for the treatment of Alzheimer’s disease, is also suitable for multiple sclerosis. Referring to our previous study in the EAE model, this is the first time we have observed the effects of HupA on animal behavior and myelination in a CPZ-induced demyelination model [23]. In this study, we have explored the molecular mechanism of HupA in promoting remyelination in both the macroscopic and microscopic aspects by animal behavior experiments, IHC, cell-based experiments, and qRT-PCR assays. It is proved that HupA can effectively ameliorate myelin regeneration at the injured site, and its mechanism may be through promoting OPCs differentiation, regulating microglia activation, and inhibiting the secretion of pro-inflammatory cytokines. These results provide an important reference for the clinical treatment of demyelinating diseases by promoting OPC differentiation and remyelination, which are expected to completely cure MS soon. However, its specific mechanism remains to be further deciphered. Thus, the subsequent research will also focus on the possible signaling pathways affected by HupA and how it can be used to promote OPC differentiation and inhibit microglial activation. Besides, there are some unresolved issues, including the effect of long-term use of HupA on chronic diseases and its possible side effects. In general, complete clinical trials are required for the evaluation and verification of the clinical efficacy of HupA. The above studies provide new ideas for further exploring the drugs against MS and other neurological diseases in the future.

## 4. Materials and Methods

### 4.1. Animals 

Sixty-four male C57BL/6 mice (five-week-old) were purchased from Shanghai Slack Laboratory Animal Co., Ltd. Mice were kept in the animal installation retained at a 12 h/12 h light-dark cycle, 20–22 °C, and 50–60% humidity, with adequate water and food. Mice were allowed to adapt to the new environment for at least one week before the experiments. All animal procedures were consistent with the Fudan University and National Institutes of Health Guide for the Care and Use of Laboratory Animals (NIH publication #85-23).

### 4.2. HupA Treatment

Cuprizone (Sigma-Aldrich, 0.3% (*w*/*w*)) was combined with rodent chow (Trophic Animal Feed High-Tech Co.,Ltd, Nantong, China.) to feed them when mice were seven weeks old (54). HupA (Pizer, Montreal, QC, Canada) was dissolved in dimethyl sulfoxide (DMSO) and diluted to a concentration of 0.2 mg/kg. Mecamylamine (Sigma-Aldrich, St. Louis, MO, USA, 10 mg/kg), a nicotinic acetylcholine receptor (nAChR) antagonist, was injected 30 min before administration of HupA. The mice were then randomly divided into different groups (n = 16 for each group). In the first group, mice received regular chows every day. However, mice in other groups were fed rodent chows containing 0.3% CPZ for 5 weeks with either water (CPZ + Veh) or HupA 0.2 mg/kg/day (CPZ + HupA) or HupA 0.2 mg/kg/day, mecamylamine 10 mg/kg/day (CPZ + HupA + Mec). From the fourth week, HupA and mecamylamine were injected intraperitoneally into the mice at a volume of 0.2 mL/mouse for 2 weeks. The mice’s weights were measured every four days during the study. Behavioral tests were conducted at weeks 0, 5, and 6 to assess motor coordination, anxiety-like behavior, and working memory.

### 4.3. Animal Behavior

The BWT, rotarod test, Y maze, and OFT were conducted at weeks 0, 5, and 6. Only one test was executed every day in the following order: (1) OFT; (2) Y maze; (3) Rotarod test; (4) BWT.

### 4.4. BWT and Rotarod Test 

BWT and rotarod tests can assess the coordination and balance ability of the rodents. In the beam walking test, the balance (1 m × 1 cm) beam was placed at the height of 50 cm, and a safety box was set at the end of the balance beam. The safety box was filled with bedding, and the mice were trained to enter the safety box through the balance beam one day in advance. The passing time was recorded accurately with a stopwatch. 

Before the rotarod test, the mice were trained to maintain balance and exercise on the rotary rod instrument (YLS-4C, Shanghai Bio-will Co., Ltd, China) twice a day at a low speed of 5 r/min for one day. In the test, the instrument was turned on, and the basic parameters were set at an initial speed of 4.0 rpm, acceleration of 0.1 rpm/s, and duration of 300 s. The mice were thereafter placed on the rotarod to start the experiment. Finally, the dwell time on the rotarod was recorded.

### 4.5. Y Maze and OFT

We used the Y maze apparatus to measure the working memory by noting spontaneous alternation. Rodents are naturally inclined to explore new environments [79]. Normal mice can memorize the arm they have explored and enter the other arm of the maze. Mice were allowed to stay at the end of one arm and explore all the other arms for 8 min. We named three arms A, B, and C, respectively. Alternations defined as overlapping entry sequences (such as ABC, BCA) were counted as a percentage: alternation% = (alternation times)/(total number of arm entries − 2) × 100 [80].

OFT can be used to analyze autonomous activity and explore the behaviors of rodents under novel circumstances. The OFT system consists of a computer, a video camera, and a plastic cage (Length 40 × Width 40 × Height 50 cm). Mice were allowed to move freely in the cage and probe the space for 5 min. The video camera was placed above the square to record their movement. Data were obtained by super maze software, an automatic animal behavior recording system, such as total distance, center distance, etc.

### 4.6. Tissue Preparation

All mice were anesthetized with chloral hydrate (300 mg/kg, i.p.) and sacrificed at the end of week 6 (day 42). The mice brains (n = 12 mice peer group) were carefully extracted after the saline infusion and fixation with 4% paraformaldehyde (PFA). Then brains were soaked in 30% sucrose solution for at least 24 h and finally preserved at −80 °C for IHC (n = 4) and TEM (n = 4). After giving saline infusion, other mice (n = 8 mice peer group) brains were removed, and corpus callosum tissue was found under a microscope. Then the corpus callosum was quickly resected on ice with fine forceps. These samples were preserved at −80 °C for western blot assays (n = 4) and qRT-PCR (n = 4).

### 4.7. Western Blot

Protein extract of the corpus callosum (20 µg) was separated on sodium dodecyl sulfate-polyacrylamide gel electrophoresis (SDS-PAGE) and then electrophoretically transferred to the nitrocellulose membrane (Millipore, Burlington, MA, USA). Electrophoretic separation was performed on 12% separating gel and 5% stacking gel. After blocking with 5% fat-free milk at room temperature for 1 h, the membrane was incubated overnight with rabbit anti-MBP and β-actin (1:1000, Abcam, Cambridge, UK) antibodies at 4 °C. Subsequently, the membrane was incubated with HRP-conjugated goat anti-rabbit secondary antibodies (1:10,000, Abcam, Cambridge, UK) at room temperature for 2 h. Immunoblotting was performed with a chemiluminescence system, and the data were analyzed by Quantity One Software (Bio-Rad, Hercules, CA, USA).

### 4.8. OPC Isolation and Culture

Fifteen postnatal (1-day-old) SD rats were purchased from Shanghai Slack Laboratory Animal Co. Ltd., and primary OPCs were obtained from rats. The cerebral cortex was dissected under aseptic conditions and placed in the sterile phenol-free Dulbecco’s modified Eagle’s medium (DMEM, 10% FBS, 4 mM L-Glutamine, 1 mM sodium pyruvate, Sigma), which contained 100 U/mL penicillin and streptomycin. A Pasteur pipette was used to dissociate the cells into a single-cell suspension, which was inoculated in a T75 cell culture bottle for the culture. After 8–10 days, the OPC proliferation medium was added, and the mixture was placed in the cell shaker (200 rpm) for overnight shaking. The next day, OPCs were inoculated into 24-well plates at a density of 4 × 10^5^ cells/well. Thereafter, the OPC differentiation medium with HupA (0.1/1/10 μM) was added daily for six consecutive days, and ICC was performed to observe OPC differentiation.

### 4.9. OPC Differentiation and CTG

OPCs were inoculated into 96-well plates at a density of 2 × 10^4^ cells/well. Then 100 μL OPC differentiation medium with HupA (0.1/1/10 μM) was added to the cells. After incubation for 72 h, 30 μL/well CTG was dripped in and tested with an enzyme marker.

#### 4.9.1. ICC

After 6 days of HupA intervention in OPCs, the medium was sucked away and washed twice with PBS. Then the paraformaldehyde was fixed at 4 °C for half an hour. At 4 °C, OPCs were sealed with 1% immune tissue blocking fluid for overnight and subsequently, incubated overnight with rabbit anti-MBP (1:100, Abcam, Cambridge, UK) at 4 °C. Then OPCs were incubated with the corresponding secondary antibody at room temperature for 2 h. Finally, OPCs were photographed with a common fluorescence microscope (Leica DMI6000).

#### 4.9.2. IHC

Mice brains were cut into several brain slices (25 µm) by microtome. At room temperature, the brain slides were sealed with 1% immune tissue blocking fluid for 2 h and incubated overnight with rabbit anti-MBP (1:100, Abcam, Cambridge, UK), anti-Iba-1 (1:100, Wako, Japan), anti-CC1 (1:100, Millipore, Burlington, MA, USA), goat anti-Olig2 (1:100, Millipore, Burlington, MA, USA), anti-PDGFRα (1:100, Millipore, Burlington, MA, USA) respectively at 4 °C. Subsequently, the brain slides were incubated with the corresponding secondary antibody at room temperature for 1 h. Then, the corpus callosum site of slices was photographed with a multiphoton laser scanning confocal microscopy system (Olympus Fluoview FV1000) or common fluorescence microscope (Leica DMI6000).

#### 4.9.3. LFB Staining

In order to evaluate demyelination within the corpus callosum, brains were paraffin-embedded, and paraffin sections were stained with hematoxylin and LFB. Paraffin tissues were placed in LFB/cresyl violet overnight at 55 °C and washed with double distilled water and 95% ethanol to facilitate the removal of excess dye. The gray matter in the lithium carbonate solution was completely separated from the white matter of the sections, followed by washing in 75% ethanol and double distilled water.

### 4.10. TEM

The corpus callosum was fixed with 3% glutaraldehyde and 1% osmic acid and washed with 0.1 PBS. Thereafter, ultrathin slides were prepared by a Reichert ultramicrotome with a diamond-tipped knife, contrasted with an ultra-microtome. The ultrastructure was then photographed by Philips CM120 electron microscope at Shanghai Medical College of Fudan University.

### 4.11. Microglia Isolation and Quantitative RT-PCR (qRT-PCR)

Combined with physical separation and digestion by DNase I, the whole mice’s brains were prepared into a single-cell suspension. Then microglia were separated by percoll solution with different concentrations to enrich microglia and separate them from myelin and red blood cell fragments. Next, CD11b+ cells were separated by immunomagnetic beads to improve the isolation purity of cells, and microglia were isolated.

The relative expression of pro-inflammatory (*iNOS/CD16/IL-1β/IL-18/TNF-α*), anti-inflammatory (*Arg1/CD206*) transcription factors, and markers in the microglia was analyzed by qRT-PCR. Total RNA was extracted from microglia and isolated with Trizol (Invitrogen, USA). The relative abundance of target mRNAs was quantified by SYBR Green qRT-PCR (Hifair^®^ III real-time PCR detection system, Yeasen, China). The primer sequences of each target mRNA were as follows:

*GAPDH*-Forward: 5′-CCTTCCGTGTTCCTACCC-3′,

*GAPDH*-Reverse: 5′-GCTTCACCACCTTCTTGATGT-3′

*iNOS*-Forward: 5′-GTTCTCAGCCCAACAATACAAGA-3′

*iNOS*-Reverse: 5′-GTGGACGGGTCGATGTCAC-3′

*Arg1*-Forward: 5′-CTCCAAGCCAAAGTCCTTAGAG-3′

*Arg1*-Reverse: 5′-AGGAGCTGTCATTAGGGACATC-3′

*CD16*-Forward: 5′-TGCTTTTGCAGACAGGCAGA-3′

*CD16*-Reverse: 5′-GAGTTCCCAGGGTTGTGGGT-3′

*CD206*-Forward: 5′-CTCTGTTCAGCTATTGGACGC-3′

*CD206*-Reverse: 5′-CGGAATTTCTGGGATTCAGCTTC-3′

*TNF-α*-Forward: 5′-GAACTCCAGGCGGTGCCTAT-3′

*TNF-α*-Reverse: 5′-TGGTGGTTTGTGAGTGTGAGG-3′

*IL-1β*-Forward: 5′-TGCCACCTTTTGACAGTGATGA-3′

*IL-1β*-Reverse: 5′-TGCCTGCCTGAAGCTCTTGT-3′

*IL-18*-Forward: 5′-TTCTCCCCTGTGGTGTGCTG-3′

*IL-18*-Reverse: 5′- CCACAGAGAACCCCCACCAG-3′

*Gapdh*, a housekeeping gene, was used as an internal reference for analysis standardization. N-fold differential expression was measured by the 2^−ΔΔCt^ method for the relative quantification and expressed as a percentage of *Gapdh*. A melting curve was used to determine the purity of the amplified band. qRT-PCR products were also sequenced to confirm their identities.

### 4.12. Statistical Analysis

G-ratio was analyzed for the myelination in electron microscopic photomicrographs by using ImageJ software. One-way and two-way ANOVA with Benjamini, Krieger, and Yekutieli post hoc tests were used to analyze the differences between groups by GraphPad Prism version 8.0 (GraphPad Software, San Diego, CA, USA). The data have been represented as mean ± SEM. Each experimental group had at least four mice. The value of *p*< 0.05 was regarded as statistically significant.

## 5. Conclusions

HupA improved impaired motor coordination, spatial working memory, and anxiety-like behaviors in demyelinated mice.

HupA did not affect the proliferation of OPCs but promoted their differentiation and remyelination.

HupA inhibited the activation of microglia in demyelinating mice. In addition, the mRNA expression of pro-inflammatory and anti-inflammatory microglia-related genes in the corpus callosum was down-regulated and up-regulated, respectively.

## Figures and Tables

**Figure 1 ijms-23-16182-f001:**
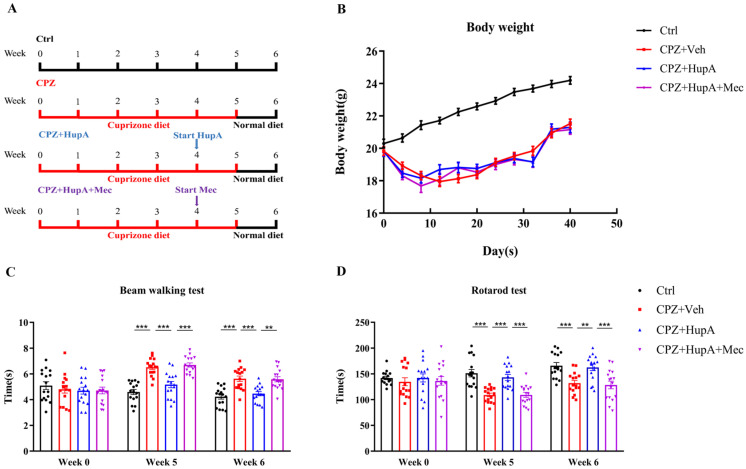
HupA improved motor coordination function in CPZ-induced demyelination. Mice were administered chow containing 0.3% CPZ until week 5 and were intraperitoneally injected with HupA and mecamylamine for 2 weeks of treatment from week 4 to 6 of CPZ feeding. (**A**) Administration of CPZ, HupA, and mecamylamine. (**B**) body weight change. (**C**) BWT. (**D**) rotarod test. The results represent the mean ± SEM (** *p* < 0.01, *** *p* < 0.001, n = 16 mice per group).

**Figure 2 ijms-23-16182-f002:**
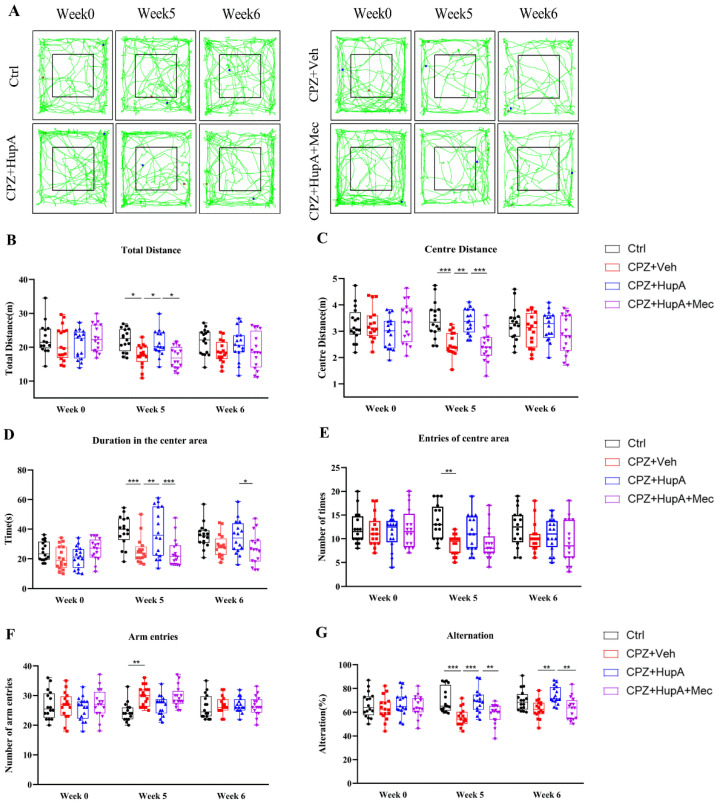
HupA improved anxiety-like symptoms and enhanced memory in demyelinated mice. (**A**) Representative images demonstrated typical cases of exploratory behavior in OFT among the four groups, containing total distance (**B**), center distance (**C**), duration of the movement in the central area (**D**), and entries of center area (**E**) in open field test during 5 min. All the mice were tested in a Y maze test for 8 min, including arm entries (**F**); spontaneous alternative (**G**). The results represent the mean ± SEM (* *p*< 0.05, ** *p* < 0.01, *** *p* < 0.001, n = 16 mice per group).

**Figure 3 ijms-23-16182-f003:**
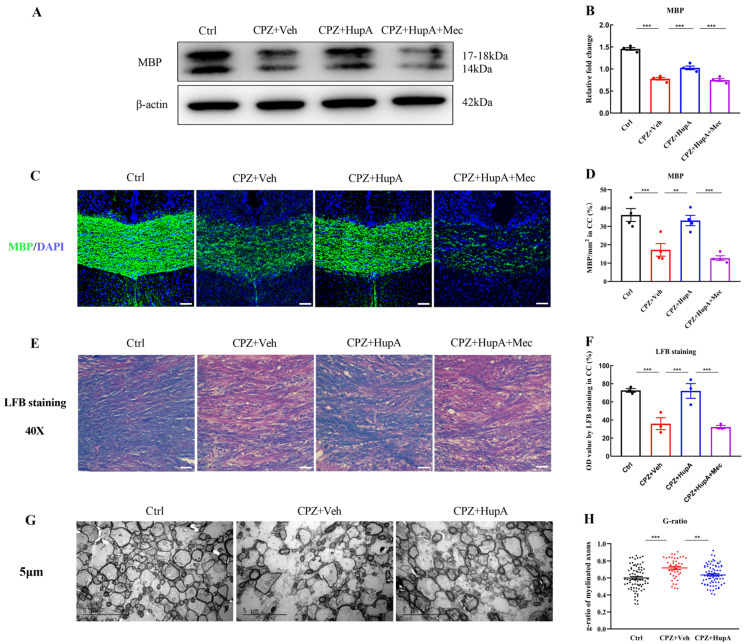
HupA promoted the formation of the new myelin sheath in CPZ-induced demyelination. (**A**–**F**) Histological evaluation of the demyelination was performed by western blot, MBP, and LFB staining and quantified by Image-J software, scale bar = 50 μm. Representative images of western blot, LFB, and MBP staining were observed in the corpus callosum of the brain. (**G**) TEM ultrastructure of the corpus callosum (Scale bar = 5 μm). (**H**) Each group of g-ratio in the corpus callosum. The results represent the mean ± SEM (** *p* < 0.01, *** *p* < 0.001, n = 3/4 mice per group).

**Figure 4 ijms-23-16182-f004:**
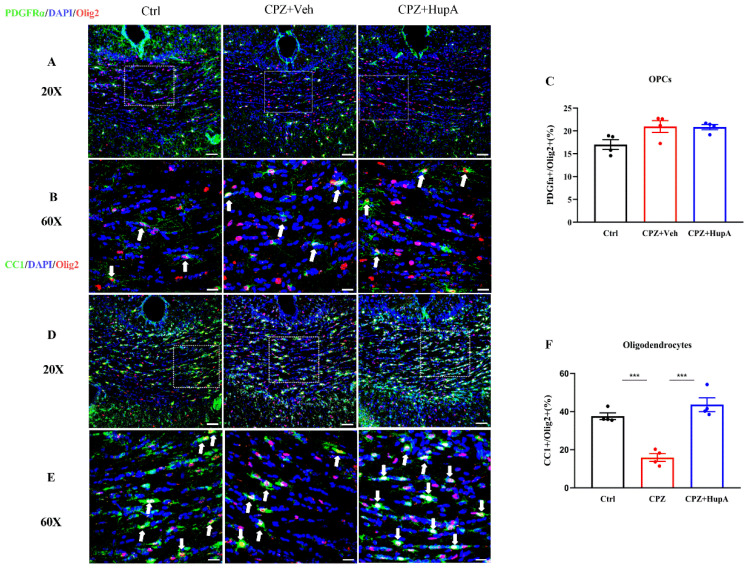
HupA did not display any effect on the amount of PDGFRα^+^ Olig2^+^ OPCs but increased the amount of CC1^+^ Olig2^+^ mature oligodendrocytes in the corpus callosum. (**A**) Double IHC with anti-PDGFRα and anti-Olig2 in the corpus callosum. PDGFRα (green), Olig2 (red), DAPI (blue), scale bar = 50 μm. (**B**) Partially enlarged view of figure A. White arrows represents OPCs, scale bar = 50 μm. (**C**) The percentage of PDGFRα^+^/Olig2^+^ OPCs in the corpus callosum of each group. (**D**) Double IHC with anti-CC1 and anti-Olig2 in the corpus callosum. CC1 (green), Olig2 (red), DAPI (blue), scale bar = 50 μm. (**E**) Partially enlarged view of figure D. White arrows represent the mature oligodendrocytes, scale bar = 50 μm. (**F**) The percentage of CC1^+^/Olig2^+^ mature oligodendrocytes in the corpus callosum of each group. The results represent the mean ± SEM (*** *p* < 0.001, n = 4 mice per group).

**Figure 5 ijms-23-16182-f005:**
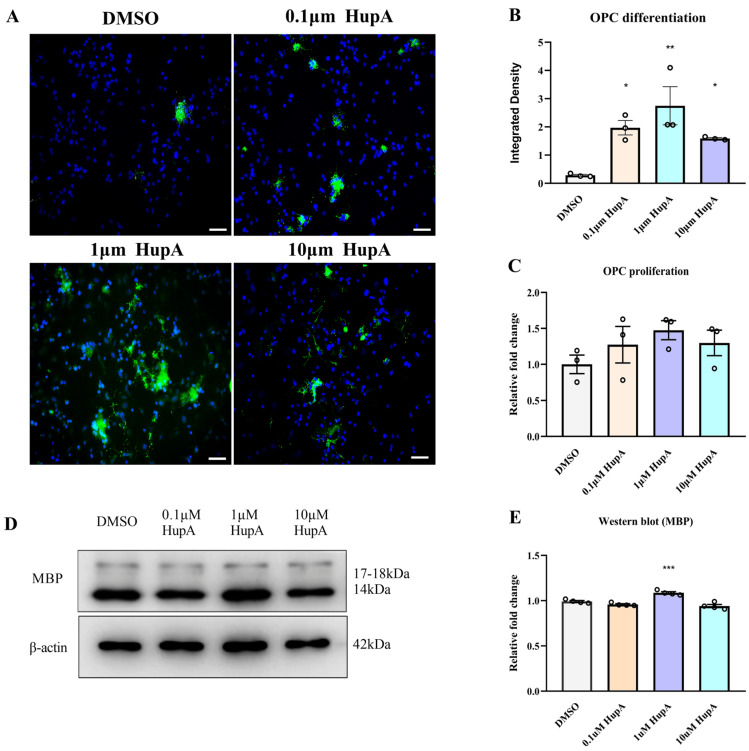
HupA treatment induced OPC differentiation into mature oligodendrocytes in vitro. (**A**,**B**) ICC with anti-MBP of OPCs cultured in vitro. MBP (green), DAPI (blue), scale bar = 50 μm. (**C**) The result of OPC proliferation by CTG. (**D**,**E**) Western blot was used to test MBP expression and analyze the potential effect of drugs on OPC differentiation. The results represent the mean ± SEM (All *p* values were compared with the control group, * *p*< 0.05, ** *p* < 0.01, *** *p* < 0.001, n = 3/4 mice per group).

**Figure 6 ijms-23-16182-f006:**
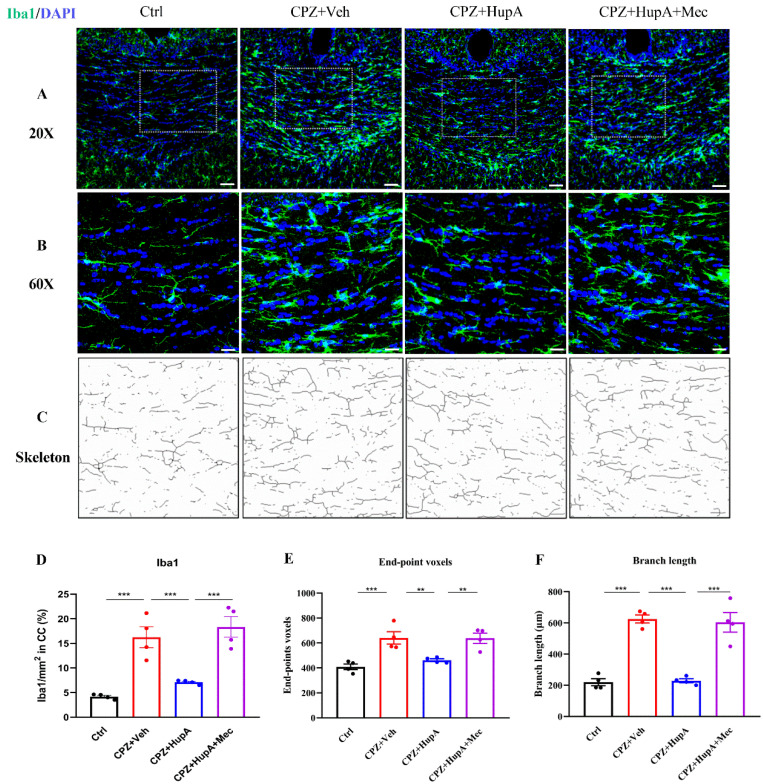
HupA treatment reduced the quantity of microglia in the corpus callosum. (**A**,**B**) Iba1 IHC. (**C**) Morphological analysis of the structure of microglia. (**D**–**F**) Quantitative analysis of the proportion of microglia in the corpus callosum, the number of branch ends, and total branch length. The results represent the mean ± SEM (** *p* < 0.01, *** *p* < 0.001, n = 4 mice per group).

**Figure 7 ijms-23-16182-f007:**
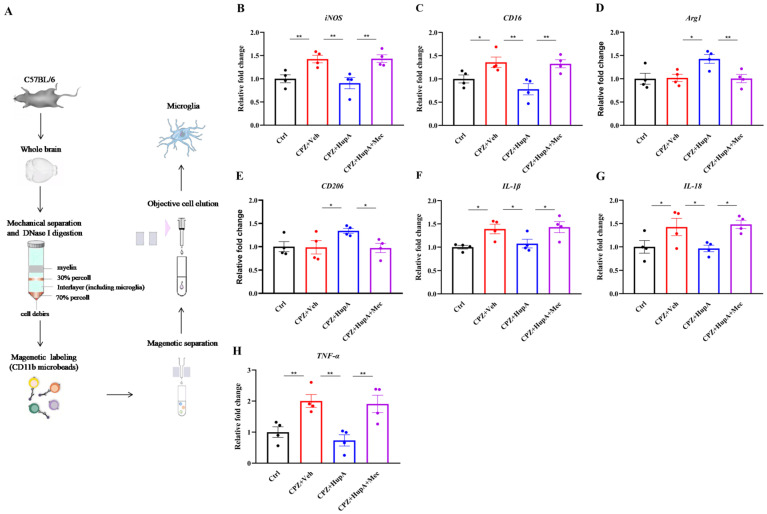
HupA modulated the expression of pro-inflammatory and anti-inflammatory molecules in the microglia. (**A**) Administration of microglia extraction by magnetic beads. (**B**–**E**) The result of qRT-PCR was to assess the mRNA expression of pro-inflammatory genes (*iNOS* and *CD16*) and anti-inflammatory genes (*Arg1* and *CD206*) in the microglia. (**F**–**H**) The result of qRT-PCR analysis to detect the expression of pro-inflammatory cytokines (*IL-1β*, *IL-18*, and *TNF-α*) in the microglia. The results represent the mean ± SEM (* *p* < 0.05, ** *p* < 0.01, n = 4 mice per group).

## Data Availability

The original contributions provided in the study are included in the article/Appendix A. Further inquiries can be directed directly to the corresponding author.

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
