# Peer review of "Huperzine—A Improved Animal Behavior in Cuprizone-Induced Mouse Model by Alleviating Demyelination and Neuroinflammation"

_ijms, 2022, doi:10.3390/ijms232416182_

Round 1

Reviewer 1 Report

Dear authors,

The authors decipher the role of hupA as a neuroprotector in MS model following previous data. It is an interesting work, but as a major concern, in mouse analyses, differences between groups should be assessed by two-way ANOVA with Benjamini, Krieger and Yekutieli post hoc test. Authors should run a new statistical analysis of their data.

In details:

2.1 HupA enhanced the motor coordination function of demyelinated mice

Authors wrote: “However, there was no difference observed between HupA treatment and control mice (Figure 1B).” line 78-79. In fact following the figure legend I believe the authors are referencing control mice when they should say “CPZ+Veh”

2.2. HupA diminished anxiety and ameliorates spatial memory deficit in demyelinated mice

There is no effect on total distance but indeed on the different ways to measure the “time” spent in the centre area, which are: 1. centre distance, 2. duration of the movement in the centre area and 3. entries of centre area. CPZ treatment seems to have effects only at week 5, but not at week 6 (if I am mistaken, then the authors should add the Pvalue on week 6). Then HupA treatment counteracts CPZ treatment for centre distance and duration of the movement in the centre area but not for entries of centre area. Again, if I am mistaken, then the authors should add the Pvalue. These effects are only apparent at week 5 but not at week 6, so effects are observed only after 1 week of treatment and not 2 as claimed by the authors (line 105).

I wonder if Control group and CPZ-HupA group have statistically “un-different phenotype”. Results of statistical analysis between the 2 groups should be run and discussed.

Also, it looks like BWT, Rotarod and Y maze phenotypes are maintained from week 5 to week 6, but not for OFT. It should be discussed.

2.3. HupA promoted the expression of myelin basic protein (MBP) in the corpus callosum and increased the quantity of the myelinated axons.

- it is unclear when the mice were sacrificed for Western Blot analysis as well as for MBP and LFB staining (5 or 6 weeks?). TEM appeared to be performed at 6 weeks protocol. Authors should be clear in the text about that.

- Western blot seems conclusive but I have doubts on the MBP IF results. First, the pictures in figure 3C are not representative of the graph (figure 3D). The picture let us think there is an almost reversion of CPZ treatment with HupA but the graph does not. Also the authors base their claim on one IF on 4 different mice. Measurement should be performed on different sections of the same animal (more than 3) and on different animals. Technical repetitions should be accounted. Finally, the most representative picture of the results should be selected.

- The choice of MBP and LFB as markers for myelinization should be explained clearly and referenced for the readers.

- The authors claim:As shown in the picture, the average myelin sheath thickness was increased 140 markedly. The g-ratio of HupA-treated mice was significantly reduced, thus suggesting that HupA 141 might also promote the formation of new myelin sheath” (line 140-142). At this point, I do not understand how the results can discard the possibility that hupA only protect from demyelination. To be conclusive, I wonder if the authors should compare TEM from CPZ mice at week 4 with CPZ mice at week 6 and CPZ-HupA week 6. This might give some input in vivo . It should be considered.

2.4. HupA did not have impact on the quantity of OPCs, but increased the quantity of the mature oligodendrocytes in vivo settings

High number of cells were counted even though only from one IF from 4 mice. Results look conclusive.

2.5. HupA did not affect the proliferation of OPCs, but promoted the differentiation of OPCs in 174 vitro settings

- In figure 5A, it looks like a lot of floating dead cells… also the CTG results show high SEM. Even though the authors tend to show no variation in proliferation rate I am not convince by the results. OPC proliferation test: it looks like N=6 but there is no references on biological repetition. Are they 6 different extraction at different day? 6 different rats ? Are they simply technical repetitions? This need to be clear, it should be biological repetitions at different days from different rats.

2.6. HupA can regulate the activation of microglia in the corpus callosum of demyelinated mice

Once again, analysis is based on 4 mice, but only one IF per mouse. Also there is no indication on how the authors determined numbers of end point and branch length, neither references.

2.7. HupA down-regulated and up-regulated the mRNA expression of pro-inflammatory and anti-inflammatory microglia related genes in the corpus callosum respectively

The CPZ-HupA-Mec condition is not always present in the results, missing from the TEM analysis, proliferation/differentiation (Figure 4) amd mRNA analysis (Figure 7). This should be complemented or discussed.

Minor concerns:

BWT is used throughout the results but its meaning is only apparent in methods (line 358). Meaning of acronym should be given the first time the words are used. Same for OFT (line 96), LFB (line 127), OPC is not specified (line 152), neither CTG (line 178)            .

Y-maze, Y always in capital letter.(line 279)

Dose for Mecamyline is shown in the text, you should do the same for the others drugs, or only mentioned it in methods.

Line 105, a verb is missing

References:

Original references are missing, notably when describing the long-known phenotype of CPZ induced model. 2 recent reviews should be cited or should be used by the authors to correctly references motor coordination, balance (line 80), anxiety (line 96), cognitive impairment in demyelination lesion (line 107) etc… Review cuprizone: Molecular Neurodegeneration https://pubmed.ncbi.nlm.nih.gov/35526004/ current neuropharmacology: https://www.ncbi.nlm.nih.gov/pmc/articles/PMC6343207/

Figure:

-          Meaning Pvalue should be presented in figure legend and not in the text (line 83, line 84, figure 1 for example)

-          Figure 3 : “LFB staining” should be added next to figure 3E (the same way MBP was mentioned nect to figure 3C)

Methods:

-          OFT: the authors should provide more information on what or how the data is obtained

-          CTG: reference missing

Discussion:

In my opinion, several parts belong to introduction. Also, I believe that the authors did not discuss the role of mecamylamine in their experiments.

Reviewer 2 Report

The manuscript by Zhang et al examines the effects of the drug Huperzine A, an acetylcholinesterase inhibitor, on the histopathological and behavioral consequences of demyelination in the cuprizone model of multiple sclerosis. In HupA-treated demyelinated mice, authors claimed to have significant improvements in motor coordination, anxiety, and memory. The authors further show that HupA promotes myelination by promoting the differentiation of OPCs. HupA also affected microglial activation and transcriptional regulation of their pro and anti-inflammatory genes. The study seems interesting and important however data is neither compelling nor sufficient to endorse the claims made by the authors.  

I have the following major comments:

1.       Figure 2 page 4, despite continuous treatment with HupA, why mice did not continue to show improved outcomes at week 6? In C-E, on what basis authors determine the size of the central area to analyze? It appears that the total distance covered (A) does not show any change and authors seem to choose the central area arbitrarily.

2.       Fig. 2 F-G, authors have magnified the scale bars to enhance the signal, it is not a standard way to present data. All scales should start from 0. The differences in F and G are so small and are visible because authors enhanced the scale bar. This marginally visible though statistically significant data is not compelling. G is in even percent but actual data values which further questions the validity of data.   

3.       Figure 4 C has the same scale issue.

4.       Figure 3G: Images lack sufficient resolution to critically analyze the authors claims.  G has two panels of images with two different scales. What do they signify?

5.       Fig 5 D: Again, scale issue. Due to this issue, the marginal effect of 10-15% misleads to appear robust. In fig 5A-B; the authors do not show any proliferation assay to conclude that proliferation is not occurring. Similarly, the effect in differentiations is so small and n appears underpowered making the overall data not convincing.

6.       Results 2.7 lines 217-218, Authors claimed HupA effect on microglial gene upregulation. There is not sufficient data to claim that. There are all kinds of cells in the tissue contributing to the changes in gene expression. Unless authors confirm the microglial overexpression by appropriate methods such as qPCR from isolating microglia or by RNAscope, their claims are not convincing.

7.       Similarly, the discussion (page 10 lines 268-271) in the manuscript claims microglial changes as the main effector of HupA’s effect; the data is not sufficient to claim so.   

8.       The authors used only male mice in this study. The rationales must be explicitly stated.

9.       Authors should also elaborate on how they isolated corpus callosum, any microdissection?

10.   Data/images should be presented from OPC cultures stained with appropriate markers to confirm the cell type.

Minor comments:

1.       Authors have not mentioned the full forms of abbreviations in the manuscript text.

2.       Several statements do not make sense for example page 2 line 77, Page 3 line 107.

Round 2

Reviewer 1 Report

I thank the authors for taking into account my critics. I understand that their previous articles were accepted with low “N” (I refer to single IF on 4 mice). I am not used to it, and I believe that even though other methods tend to support their conclusions, the same rigorous methods should be asked for each experiment. Nonetheless I will let the editor take the final decision.

But the authors claimed that they will provide as soon as possible the results of new in vitro culture experiments as their previous results were not convincing (remarks from myself but also from reviewer 2).

I will wait for the results.

Reviewer 2 Report

The authors have significantly improved the manuscript however have not provided the requested data from the first revision yet. Without the requested data the claims in the manuscript can not be endorsed for publication. 

Round 3

Reviewer 1 Report

Dear authors,

Thanks for the modifications you brought. I have only suggestion to the authors in order to facilitate the understanding for the future readers of this work

I would like to suggest you change the title for figure 5E (as it is the same as fig5B, it is quite confusing)... maybe "MBP". 

Also, I suggest you change the color code for the increasing concentration for hupA in figure 5 as it is the same code color used for the other graph where it is representing the different injection condition in the in vivo.

Thank you for your work. 

Reviewer 2 Report

In the revised manuscript authors addressed all comments and provided compelling data. I have no more comments.  

Author Response

Dear editor and reviewer 2:

 We would like to thank the editors and reviewers for your patience. We are very sorry for keeping you waiting so long. Thanks again for your reply, recognition and hard work.

Thank you for your careful review. We appreciate your positive evaluation of our work and agree with the comments regarding the limitations of our study. We wish good health to you, your family, and community.

Sincerely

Jun Wang